# A voice-based biomarker for monitoring symptom resolution in adults with COVID-19: Findings from the prospective Predi-COVID cohort study

**Guy Fagherazzi**[1]*, **Lu Zhang**[2], **Abir Elbéji**[1], **Eduardo Higa**[1], **Vladimir Despotovic**[3], **Markus Ollert**[4,5], **Gloria A. Aguayo**[1], **Petr V. Nazarov**[2,6], **Aurélie Fischer**[1]

**1** Deep Digital Phenotyping Research Unit. Department of Precision Health, Luxembourg Institute of Health, 1 A-B rue Thomas Edison, L-1445 Strassen, Luxembourg, **2** Bioinformatics Platform, Luxembourg Institute of Health, 1A-B, rue Thomas Edison, L-1445 Strassen, Luxembourg, **3** Department of Computer Science, Faculty of Science, Technology and Medicine, University of Luxembourg, Avenue de la Fonte 6, L-4364 Esch-sur-Alzette, Luxembourg, **4** Department of Infection and Immunity, Luxembourg Institute of Health, 29, Rue Henri Koch, L-4354 Esch-sur-Alzette, Luxembourg, **5** Department of Dermatology and Allergy Center, Odense Research Center for Anaphylaxis, University of Southern Denmark, 5000 Odense, Denmark, **6** Multiomics Data Science, Luxembourg Institute of Health, 1A-B, rue Thomas Edison, L-1445 Strassen, Luxembourg

* guy.fagherazzi@lih.lu

**Data Availability Statement:** Data and Code Availability: The data and code used to train and validate the models are available here: Data: https://

## Abstract

People with COVID-19 can experience impairing symptoms that require enhanced surveillance. Our objective was to train an artificial intelligence-based model to predict the presence of COVID-19 symptoms and derive a digital vocal biomarker for easily and quantitatively monitoring symptom resolution. We used data from 272 participants in the prospective Predi-COVID cohort study recruited between May 2020 and May 2021. A total of 6473 voice features were derived from recordings of participants reading a standardized pre-specified text. Models were trained separately for Android devices and iOS devices. A binary outcome (symptomatic versus asymptomatic) was considered, based on a list of 14 frequent COVID-19 related symptoms. A total of 1775 audio recordings were analyzed (6.5 recordings per participant on average), including 1049 corresponding to symptomatic cases and 726 to asymptomatic ones. The best performances were obtained from Support Vector Machine models for both audio formats. We observed an elevated predictive capacity for both Android (AUC = 0.92, balanced accuracy = 0.83) and iOS (AUC = 0.85, balanced accuracy = 0.77) as well as low Brier scores (0.11 and 0.16 respectively for Android and iOS when assessing calibration. The vocal biomarker derived from the predictive models accurately discriminated asymptomatic from symptomatic individuals with COVID-19 (t-test P-values<0.001). In this prospective cohort study, we have demonstrated that using a simple, reproducible task of reading a standardized pre-specified text of 25 seconds enabled us to derive a vocal biomarker for monitoring the resolution of COVID-19 related symptoms with high accuracy and calibration.

zenodo.org/record/5572855 Code: https://github.com/LIHVOICE/Predi-COVID_voice_symptoms.

**Funding:** The Predi-COVID study is supported by the Luxembourg National Research Fund (FNR) (grant number 14716273 to GF, MO), the André Losch Foundation (GF, MO), and the Luxembourg Institute of Health (GF, MO). The funders had no role in the study design, data collection and analysis, decision to publish, or preparation of the manuscript.

**Competing interests:** The authors have declared that no competing interests exist.

## Author summary

People infected with SARS-CoV-2 may develop different forms of COVID-19 characterized by diverse sets of COVID-19 related symptoms and thus may require personalized care. Among digital technologies, voice analysis is a promising field of research to develop user-friendly, cheap-to-collect, non-invasive vocal biomarkers to facilitate the remote monitoring of patients. Previous attempts have tried to use voice to screen for COVID-19, but so far, little research has been done to develop vocal biomarkers specifically for people living with COVID-19. In the Predi-COVID cohort study, we have been able to identify an accurate vocal biomarker to predict the symptomatic status of people with COVID-19 based on a standardized voice recording task of about 25 seconds, where participants had to read a pre-specified text. Such a vocal biomarker could soon be integrated into clinical practice for rapid screening during a consultation to aid clinicians during anamnesis, or into future telemonitoring solutions and digital devices to help people with COVID-19 or Long COVID.

## Introduction

The COVID-19 pandemic has massively impacted the worldwide population and the healthcare systems, with more than 200 million cases and 4 million deaths in August 2021 [1]. COVID-19 is a heterogeneous disease with various phenotypes and severity. The diversity of profiles, from asymptomatic to severe cases admitted to ICU, require tailored care pathways to treat them [2].

Except for hospitalized individuals, asymptomatic, mild, moderate COVID-19 cases are recommended to go in isolation and ensure home-based healthcare [3]. Monitoring symptom resolution or aggravation can be useful to identify at-risk individuals of hospitalization or immediate attention. An objective monitoring solution could then be beneficial, with its use potentially extended to people with Long Covid syndrome [4] to monitor their symptoms in the long run and improve their quality of life.

The pandemic has largely put under pressure entire healthcare systems, up to the point of needed national or regional lockdowns. Identifying solutions to help healthcare professionals focus on the more severe and urgent cases was strongly recommended. Digital health and artificial intelligence-(AI) based solutions hold the promise of alleviating clinicians by automating or transferring tasks that can be accomplished by the patients themselves [5]. Enabling self-surveillance and remote monitoring of symptoms using augmented telemonitoring solutions could therefore help to improve and personalize the way COVID-19 cases are handled [6].

Among all the types of digital data easily available at a large scale, voice is a promising source, as it is rich, user-friendly, cheap to collect, non-invasive, and can serve to derive vocal biomarkers to characterize and monitor health-related conditions which could then be integrated into innovative telemonitoring or telemedicine technologies [7].

Several vocal biomarkers have already been identified in other contexts, such as neurodegenerative diseases or mental health, or as a potential COVID-19 screening tool based on cough recordings [8], but no prior work has been performed yet to develop a vocal biomarker of COVID-19 symptom resolution.

We hypothesized that symptomatic people with COVID-19 had different audio features from asymptomatic cases and that it was possible to train an AI-based model to predict the presence of COVID-19 symptoms and then derive a digital vocal biomarker for easily and

quantitatively monitoring symptom resolution. To test this hypothesis, we used data from the large hybrid prospective Predi-COVID cohort study.

## Methods

### Study design and population

Predi-COVID is a prospective, hybrid cohort study composed of laboratory-confirmed COVID-19 cases in Luxembourg who are followed up remotely for 1 year to monitor their health status and symptoms. The objectives of Predi-COVID study are to identify new determinants of COVID-19 severity and to conduct deep phenotyping analyses of patients by stratifying them according to the risk of complications. The study design and initial analysis plan were published elsewhere [9]. Predi-COVID is registered on ClinicalTrials.gov (NCT04380987) and was approved by the National Research Ethics Committee of Luxembourg (study number 202003/07) in April 2020. All participants provided written informed consent to take part in the study.

Predi-COVID includes a digital sub-cohort study composed of volunteers who agreed to a real-life remote assessment of their symptoms and general health status based on a digital self-reported questionnaire sent every day for the first 14 days after inclusion, then once a week during the third and fourth weeks and then every month for up to one year. Participants were asked to answer these questionnaires as often as possible but were free to skip them if they felt too ill or if symptoms did not materially change from one day to the other.

Predi-COVID volunteers were also invited to download and use, on their smartphone, Colive LIH, a smartphone application developed by the Luxembourg Institute of Health to specifically collect audio recordings in cohort studies. Participants were given a unique code to enter the smartphone application and perform the recordings.

Data collection in Predi-COVID follows the best practices guidelines from the German Society of Epidemiology [10]. For the present work, the authors also followed the TRIPOD standards for reporting AI-based model development and validation and used the corresponding checklist to draft the manuscript [11].

In the present analysis, we included all the Predi-COVID participants recruited between May 2020 and May 2021 with available audio recordings at any time point in the first two weeks of the follow-up and who had filled in the daily questionnaire on the same day as the audio recordings. Therefore, multiple audio recordings were available for a single participant.

### COVID-19 related symptoms

Study participants were asked to report their symptoms among a list of frequently reported ones in the literature: dry cough, fatigue, sore throat, loss of taste and smell, diarrhea, fever, respiratory problems, increase in respiratory problems, difficulty eating or drinking, skin rash, conjunctivitis or eye pain, muscle pain/unusual aches, chest pain, overall pain level (for more details, please see Table 1). We consider a symptomatic case as someone reporting at least one symptom in the list and an asymptomatic case as someone who completed the questionnaires but did not report any symptom in the list.

### Voice recordings

Participants were asked to record themselves while reading, in their language (German, French, English, or Portuguese), a standardized, prespecified text which is the official translation of the first section of Article 25 of the Universal Declaration of Human Rights of the United Nations [12] (see S1 File for more details). The audio recordings were performed in a

**Table 1. Distribution of symptoms for participants with at least one symptom reported in the 14 days of follow-up.**

| Symptom | Question | Modalities | Symptom presence (%) in symptomatic evaluations (N = 1049 symptom evaluations from 225 individuals) |
|---|---|---|---|
| Dry cough | Do you have a dry cough? | Yes/No | 44.7 |
| Fatigue | Do you feel tired? | | 50.0 |
| Sore throat | Did you have a sore throat in the past few days? | | 22.7 |
| Loss of taste and smell | Did you notice a strong decrease or a loss of taste or smell? | | 47.3 |
| Diarrhea | Do you have diarrhea? At least 3 loose stools per day. | | 7.3 |
| Fever | Do you have fever? | | 7.7 |
| Respiratory problems | Do you have respiratory problems? | | 15.3 |
| Increase in respiratory problems | Did you notice the appearance or an increase of your usual respiratory problems since the diagnosis? | | 18.9 |
| Difficulty eating or drinking | Do you have significant difficulty in eating or drinking? | | 2.7 |
| Skin rash | Did you notice any sudden-onset skin rash on the hands or feet (for example frostbite, persistent redness, sometimes painful, acute urticaria)? | | 3.5 |
| Conjunctivitis or eye pain | Did you notice the appearance of conjunctivitis or eye pain (persistent redness in the whites of the eye, itchy eyelid, tingling, burning, frequent tearing)? | | 14.8 |
| Muscle pain/unusual aches | Did you have muscle pain or unusual aches in the last days? | | 34.1 |
| Chest pain | Did you have chest pain in the last days? | | 14.6 |
| | | | **Pain level > 2 (%)** |
| Overall pain level | What is your current pain level? | Rate from 1 (low) to 10 (high) | 17.3 |

real-life setting. Study investigators provided the participants with a few guidelines on how to position themselves and their smartphones for optimal audio quality, along with a demo video.

## Pre-processing

Raw audio recordings have then been pre-processed before the training of the algorithms using Python libraries (Fig 1). First, audio files have all been converted into .wav files, using the ffmpy.FFmpeg() function by keeping the original sampling rate, i.e. 8kHz and 44.1kHz for 3gp and m4a respectively, and with 16-bit bit-depth. The compression ratio is around 10:1 for 3gp files and between 1.25:1 and 10:1 for m4a files. Audios shorter than 2 seconds were excluded at this stage. A clustering (DBSCAN) on basic audio features (duration, the average, sum and standard deviation of signal power, and fundamental frequency), power in time domain, and cepstrum has been performed to detect outliers, which were further checked manually and removed in case of bad audio quality. Audio recordings were then normalized on the volume, using the pydub.effects.normalize function, which finds the maximum volume of an audio segment, then adjusts the rest of the audio in proportion. Noise reduction was applied on normalized audios with log minimum mean square error logMMSE speech enhancement/noise reduction algorithm, which is shown to result in a substantially lower residual noise level without affecting the voice signal substantially [13]. Finally, blanks > 350 ms at the start or the end of the audio were trimmed.

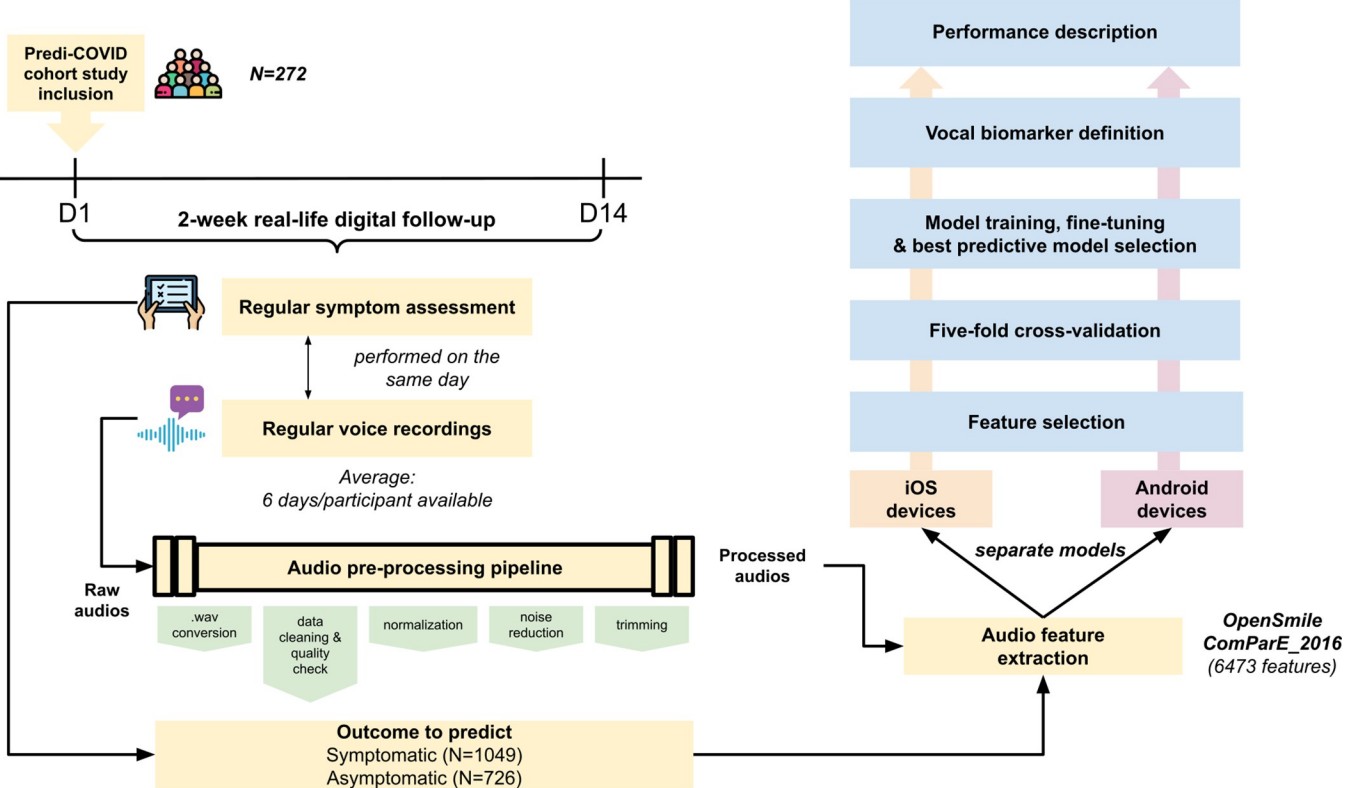

**Fig 1. General pipeline from data collection to vocal biomarker.**

## Feature extraction

We extracted audio features from the pre-processed voice recordings using the OpenSmile package [14] (see S2 File) with 8kHz for both 3gp and m4a format. We used ComParE 2016 feature set but modified the configuration file to add the Low-Level Descriptor (LLD) MFCC0, which is the average of log energy and is commonly used in speech recognition. Applying the logarithm to the computation of energy mimics the dynamic variation in the human auditory system, making the energy less sensitive to input variations that might be caused by the speaker moving closer or further from the microphone. Overall, this allowed us to extract 6473, instead of 6373 in origin, features—functionals of 66 LLDs as well as their delta coefficients.

## Data analysis

We first performed descriptive statistics to characterize the study participants included, using means, standard deviations for quantitative features, and counts and percentages for qualitative features. In each audio type (3gp and m4a), we compared the distribution of the arithmetic mean of each LLD from symptomatic and asymptomatic samples(S3 File and S4 File). Separate models were trained for each audio format (3gp/Android, m4a/iOS devices).

## Feature selection

We used recursive feature elimination to reduce the dimensionality and select meaningful information from the raw audio signal that could be further processed by a machine learning algorithm. Recursive Feature Elimination (RFE) is a wrapper-based feature selection method,

meaning that the model parameters of another algorithm (e.g. Random Forest) are used as criteria to rank the features, and the ones with the smallest rank are iteratively eliminated. In this way, the optimal subset of features is extracted at each iteration, showing that the features with the highest rank (eliminated last) are not necessarily the most relevant individually [15]. We used Random Forest as a core estimator and the optimal number of selected features was determined using a grid search in the range [100, 150, 200, 250, 300]. The optimal number of features is provided in the subsection "Best predictive model" of the "Results" section.

## Classification model selection and evaluation

We performed a randomized search with 5-fold cross-validation for the optimal hyperparameters of four frequently-used methods in audio signal processing, using their respective scikit learn functions: Support vector machine (SVM), bagging SVMs, bagging trees, random forest (RF), Multi-layer Perceptron (MLP). SVM is a widely used machine learning method for audio classification [16]. SVM constructs a maximum margin hyperplane, which can be used for classification [17]. One of the advantages of SVM is robust to the high variable-to-sample ratio and a large number of variables. A bagging classifier is an ensemble meta-estimator that fits the base classifier on random subsets of the original dataset and then aggregates their individual predictions to form a final prediction [17,18]. This approach can be used to reduce the variance of an estimator. A decision tree is a non-parametric classification method, which creates a model that predicts the value of a target variable by learning simple decision rules inferred from the data features. But a simple decision tree suffers from high variance. Therefore, we added the bagging approach described above to reduce the variance. A RF is also an ensemble learning method that fits a number of decision trees to improve the predictive accuracy and control over-fitting [19]. Different from bagging, the random forest selects, at each candidate split in the learning process, a random subset of the features, while the bagging uses all the features. We applied the random forest with different parameter configurations for the number of trees (100, 150, 200, 250, 300, 350, 400). MLP is a fully connected feedforward neural network using backpropagation for training. The scripts are made available in open source (please see the Github link below).

The performance of the models with optimal hyperparameters was assessed using 5-fold cross-validation and using the test datasets unseen by feature selection and hyperparameter tuning as well with the following indices: area under the ROC curve (AUC), balanced accuracy, F1-score, precision, recall, and Matthews correlation coefficient (MCC, a more reliable measure of the differences between actual values and predicted values) [20]. The model with the highest MCC was selected as the final model. We evaluated the significance of the cross-validated scores of the final model with 1000 permutations [21]. Briefly, we generated randomized datasets by permuting only the binary outcome of the original dataset 1000 times and calculated the cross-validated scores on each randomized dataset. The p-value represented the fraction of randomized datasets where the model performed as well or better than the original data. Calibration was then assessed by plotting reliability diagrams for the selected models using 5-fold cross-validation and by computing the mean Brier score [22]. Classification models and calibration assessments were generated using scikit-learn 0.24.2. Statistical analysis was performed using scipy 1.6.2. The plots were generated using matplotlib 3.3.4 and seaborn 0.11.1.

## Derivation of the digital vocal biomarker

For each type of device, we used the predicted probability of being classified as symptomatic from the best model as our final vocal biomarker, which can be used as a quantitative measure

to monitor the presence of symptoms. We further described its distribution, in both groups of symptomatic and asymptomatic cases and performed a t-test between the two groups.

## Results

### Study participants' characteristics

We analyzed data from N = 272 participants with an average age of 39.9 years. Among them, 50.3% were women. For recording, 101 participants used Android devices (3gp format) while 171 participants used iOS devices (m4a format). We did not observe a difference in the distribution of age, sex, BMI, smoking, antibiotic use, and comorbidity, including diabetes, hypertension, and asthma, between the two types of devices (Table 2). On average, they reported their symptoms during 6.5 days in the first 14 days of follow-up, which ended up in the analysis of 1775 audio recordings. Among them, N = 1049 were classified as "symptomatic" and N = 726 as "asymptomatic".

A total of 225 participants reported at least one symptom in the 14 days. The most observed symptoms were fatigue (50%), loss of taste and smell (47.3%), and dry cough (44.7%). Conversely, difficulty eating or drinking (2.7%) and skin rash (3.5%) were the most infrequent symptoms. The participants spoke four languages, French (44.6%), German (30.9%), English (22.6%), and Portuguese (1.9%).

### Best predictive model

We selected 100 and 250 features from 3gp and m4a audios using the Recursive Feature Elimination method (see S5 File). The optimal number of features was decided from a five cross-validated selection. For Android, 3gp format, we have observed that the selected features were mainly coming from the spectral (53%) domain, followed by cepstral (37%), prosodic (8%),

**Table 2. Overall study participants' characteristics split by type of smartphone/audio format (Predi-COVID Cohort Study, N = 272).**

| Variables [mean (SD) or N(%)] | Overall (N = 272) | Android devices / 3gp audio format (N = 101) | iOS devices / m4a audio format (N = 171) | P-values (Student's t or Chi-square) |
|---|---|---|---|---|
| **Clinical Features** | | | | |
| **Sex (% female)** | 50.3% | 46.6% | 52.4% | 0.34 |
| **Age (years)** | 39.9(13.2) | 40.1(12.5) | 39.7(13.6) | 0.80 |
| **Body mass index (kg/m$^2$)** | 25.6(4.7) | 25.6(4.3) | 25.6(4.9) | 0.97 |
| **Smoking (% current smoker)** | 15.8% | 15.5% | 15.9% | 0.75 |
| **Antibiotic use (% yes)** | 11.1% | 10.3% | 11.6% | 0.86 |
| **Diabetes (% yes)** | 2.5% | 2.3% | 2.7% | 1.00 |
| **Hypertension (% yes)** | 9.6% | 8.5% | 10.2% | 0.75 |
| **Asthma (% yes)** | 5.6% | 4.7% | 6.2% | 0.71 |
| **Voice recordings** | | | | |
| **Total audio samples available** | 1775 | 693 | 1082 | |
| "Symptomatic" labels | 1049 | 449 | 600 | <0.001 |
| "Asymptomatic" labels | 726 | 244 | 482 | |
| **Language** | | | | |
| French | 44.6% | 48.8% | 41.9% | <0.001 |
| German | 30.9% | 32.0% | 30.2% | 0.45 |
| English | 22.6% | 18.3% | 25.3% | <0.001 |
| Portuguese | 1.9% | 0.9% | 2.6% | 0.016 |
| **Number of voice recordings & assessment of symptoms per participant over 14 days** | 6.5 | 6.9 | 6.3 | |

**Table 3. Performances of the different algorithms.**

| Audio format | Model | AUC | Balanced accuracy | MCC | F1-score | F1-score 0 | F1-score 1 | Precision | Precision 0 | Precision 1 | Recall | Recall 0 | Recall 1 |
|---|---|---|---|---|---|---|---|---|---|---|---|---|---|
| **Android devices / 3gp** | **Random Forest** | 0.90 (0.92) | 0.79(0.83) | 0.60 (0.67) | 0.82 (0.85) | 0.73 (0.78) | 0.87 (0.89) | 0.82 (0.85) | 0.79 (0.80) | 0.84 (0.87) | 0.82 (0.85) | 0.68 (0.76) | 0.90 (0.90) |
| | **Support Vector Machine (SVM)** | 0.92 (0.92) | 0.83(0.83) | 0.68 (0.66) | 0.85 (0.84) | 0.79 (0.78) | 0.89 (0.88) | 0.86 (0.84) | 0.83 (0.76) | 0.87 (0.89) | 0.86 (0.84) | 0.75 (0.80) | 0.91 (0.87) |
| 449 symptomatic cases | **Bagging Tree** | 0.89 (0.91) | 0.79(0.80) | 0.62 (0.63) | 0.82 (0.83) | 0.73 (0.75) | 0.88 (0.88) | 0.83 (0.83) | 0.82 (0.81) | 0.83 (0.85) | 0.83 (0.83) | 0.66 (0.69) | 0.92 (0.91) |
| 244 asymptomatic cases | **Bagging SVM** | 0.92 (0.94) | 0.78(0.80) | 0.62 (0.65) | 0.82 (0.84) | 0.71 (0.75) | 0.88 (0.88) | 0.84 (0.84) | 0.87 (0.85) | 0.82 (0.84) | 0.83 (0.84) | 0.61 (0.67) | 0.95 (0.93) |
| | **Multi-Layer Perceptron (MLP)** | 0.88 (0.88) | 0.79(0.80) | 0.58 (0.59) | 0.81 (0.81) | 0.73 (0.73) | 0.86 (0.86) | 0.81 (0.81) | 0.74 (0.73) | 0.85 (0.86) | 0.81 (0.81) | 0.71 (0.73) | 0.87 (0.86) |
| **iOS devices / m4a** | **Random Forest** | 0.81 (0.78) | 0.72(0.70) | 0.47 (0.41) | 0.73 (0.71) | 0.67 (0.66) | 0.78 (0.74) | 0.74 (0.71) | 0.76 (0.68) | 0.73 (0.73) | 0.74 (0.71) | 0.61 (0.65) | 0.84 (0.76) |
| | **Support Vector Machine (SVM)** | 0.85 (0.84) | 0.77(0.76) | 0.54 (0.52) | 0.77 (0.76) | 0.74 (0.74) | 0.80 (0.78) | 0.78 (0.76) | 0.76 (0.72) | 0.79 (0.80) | 0.77 (0.76) | 0.73 (0.76) | 0.81 (0.76) |
| 600 symptomatic cases | **Bagging Tree** | 0.83 (0.80) | 0.74(0.73) | 0.50 (0.48) | 0.75 (0.74) | 0.69 (0.69) | 0.80 (0.78) | 0.76 (0.75) | 0.79 (0.75) | 0.74 (0.74) | 0.75 (0.75) | 0.61 (0.64) | 0.87 (0.83) |
| 482 asymptomatic cases | **Bagging SVM** | 0.86 (0.83) | 0.72(0.74) | 0.51 (0.52) | 0.73 (0.75) | 0.63 (0.68) | 0.80 (0.81) | 0.78 (0.77) | 0.88 (0.82) | 0.70 (0.73) | 0.74 (0.76) | 0.50 (0.58) | 0.94 (0.90) |
| | **Multi-Layer Perceptron (MLP)** | 0.81 (0.78) | 0.73(0.73) | 0.47 (0.46) | 0.74 (0.73) | 0.70 (0.70) | 0.77 (0.75) | 0.74 (0.73) | 0.71 (0.69) | 0.76 (0.77) | 0.74 (0.73) | 0.69 (0.72) | 0.78 (0.74) |

Two strategies were tested to assess the model performances. The first strategy was to perform a 5-fold cross-validation, in order to have more information, but potentially based on partially already seen data, the second strategy was the assessment of the model performances only on the test dataset corresponding to the training dataset where the highest performances were obtained. The numbers outside the bracket indicate the mean values from a 5-fold cross-validation across the whole dataset. In bracket is the performance on the test dataset where the model was trained on a separate training dataset. * A description of each model is available in the Material and Methods section.

and voice quality features (2%). We observed a similar trend for iOS, m4a format, as we have found that the selected features were mainly coming from the spectral (59%) domain, followed by cepstral (29%), prosodic (9%), and voice quality features (3%). For each type of device (Android, 3gp format versus iOS, m4a format), we compared the performances of each classifier (Table 3). According to the balanced accuracy and AUC criteria, we have observed that SVM models outperformed all other models. The SVM model has slightly better performance on Android devices than on iOS devices. For Android, the SVM model had an AUC of 0.92 and balanced accuracy of 0.83. The MCC was 0.68, F1-score was 0.85, precision 0.86, and recall 0.86. For iOS devices, the SVM model had an AUC of 0.85, and balanced accuracy of 0.77. The MCC was 0.54, F1-score was 0.77, precision 0.78, and recall 0.78. We can also observe in the confusion matrices that the models rarely predict the wrong symptomatic/asymptomatic status and that both mean Brier scores were low (respectively 0.11 and 0.16 for Android and iOS devices, Fig 2). The scores obtained from permuted datasets were all lower than the scores from the original dataset, i.e. the permutation p-value < 0.001.

## Vocal biomarker of symptom resolution

Based on the selected best predictive SVM models, we derived, for each type of device, digital vocal biomarkers which quantitatively represent the probability of being classified as

**Android devices / 3gp audio format**

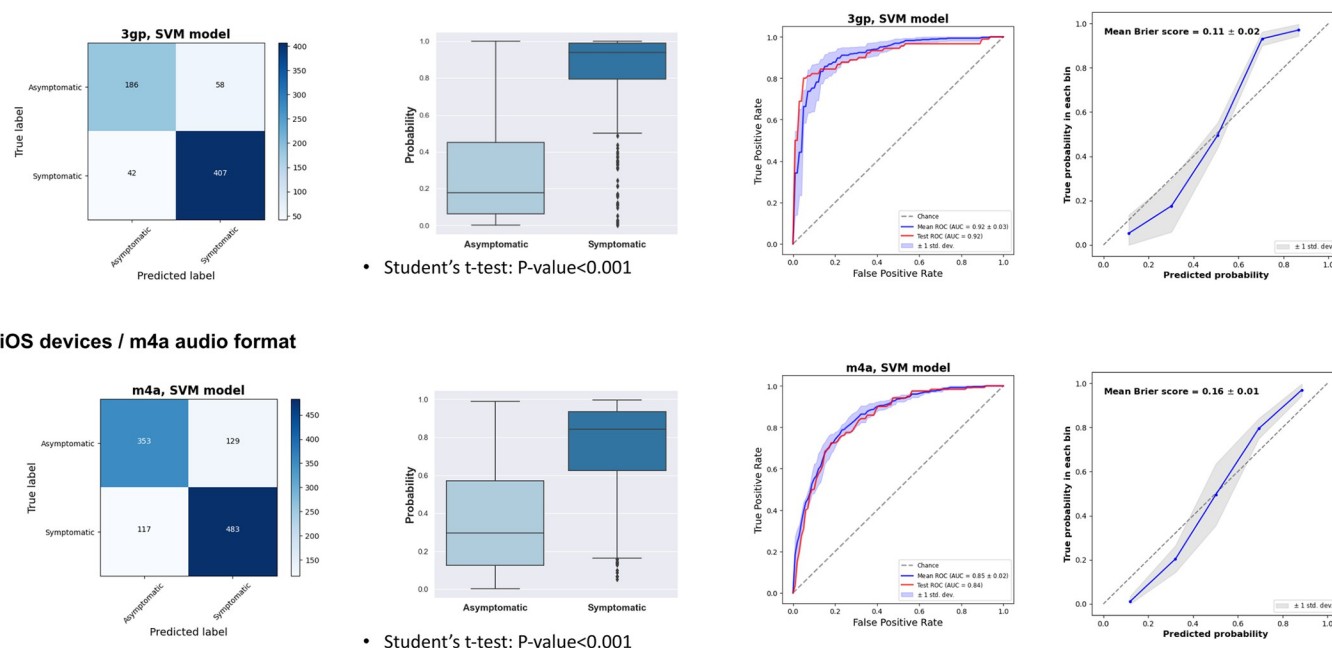

*Red lines indicate ROC for the test datasets (n = 139 and n = 216 from 3gp and m4a data respectively), where the model was trained on the separate training dataset (n = 554 and n = 866 from 3gp and m4a data respectively). Blue lines indicate the performance using 5-fold cross validation across the whole datasets. Blue areas indicate the standard deviation.

**Fig 2.** Vocal biomarker distribution in people with and without symptoms—Confusion Matrix (a), Boxplot (b), AUC (c), Calibration curve (d).

symptomatic (Fig 2). In the test set, we have observed an important difference in the distributions of the vocal biomarkers between the symptomatic and the asymptomatic categories (P<0.001 for both Android and iOS devices).

## Discussion

In the prospective Predi-COVID cohort study, we have trained an AI-based algorithm to predict the presence or absence of symptoms in people with COVID-19. We have then derived, for each type of smartphone device (Android or iOS) a vocal biomarker that can be used to accurately identify symptomatic and asymptomatic individuals with COVID-19.

### Comparison with the literature

Previous attempts to use respiratory sounds have been suggested for COVID-19 [23], but they were largely focused on the use of cough [8,24] or breathing [25] to predict COVID-19 diagnosis [26]. The most promising approach so far has been proposed by the MIT [8], as they achieved an elevated sensitivity (98.5% in symptomatic and 100% in asymptomatic individuals) and specificity (94.2% in symptomatic and 83.2% in asymptomatic individuals). Similar COVID-19 infection risk evaluation systems and large cough databases are currently under development [27], but additional research still needs to be performed to assess whether such algorithms predict the true COVID-19 status or rather the general health status of the individual [28]. We can also mention the recent work by Robotto et al, who have trained machine learning models (SVM classifiers) based on OpenSmile-derived features from voice recordings to classify individuals as either positive (n = 70), or recovered (n = 70), or healthy (n = 70)

[29]. In one sensitivity analysis, they have trained algorithms to classify positive versus recovered individuals, which is the closest approach to ours. They have shown higher performances compared to ours (AUC = 0.96 versus AUC = 0.91 and 0.85 respectively in the present work for Android and iOS devices), and have shown that similar results can be achieved regardless of the type of audio recordings used (vowel phonation, text, or cough). The slightly higher performances in their work are probably attributable to their standardized recruitment process as well as the controlled environment of the recordings, as the recording sessions were conducted in similar hospital rooms, with quiet environments and tolerable levels of background noise. In comparison to their work, our models suggest that, in real life, we can expect a slight decrease in the overall performance of the models but that it remains feasible to use such an approach without any controlled environment and relying on various devices and recording situations. No other work, i.e. focusing on individuals with COVID-19, has been reported. No comparison was possible with potential other models based on voice recordings other than coughs, nor with the objective of monitoring symptoms in people with COVID-19.

## Biological mechanisms

SARS-CoV-2 infection can cause damage to multiple organs [30,31], regardless of the initial disease severity, and can persist chronically in individuals with Long Covid [32,33]. Frequently reported COVID-19 related symptoms have now largely been described (fatigue, dyspnea, cardiac abnormalities, cognitive impairment, sleep disturbances, symptoms of post-traumatic stress disorder, muscle pain, concentration problems, headache, anosmia, ageusia [34]), and the underlying mechanisms are also described [33]. Many systems such as, but not restricted to, the respiratory, cardiovascular, neurological, gastrointestinal, and musculoskeletal systems can be altered and, if impaired, can directly impact voice characteristics. Inappropriate inflammatory response [35], increased level of cytokines such as interleukin-6 [36], stiffer and less flexible vocal cords due to inflammation, ACE2 receptors expressed on the mucous membrane of the tongue [37] or more generally a combination of central, peripheral, and negative psychological or social factors are involved both in COVID-19 pathogenesis and voice production or voice disorders. Besides, for hospitalized individuals, tracheostomy [38], intubation could also modify audio features from the voice [39]. Of note, none of the participants included in the present study underwent such a procedure.

## Strengths and limitations

This work has several strengths. First, enrolled participants were all cases who had their diagnosis confirmed by a positive PCR test, which excludes the risk of having non-infected individuals in the asymptomatic category, and false-positive individuals in the symptomatic group. The prospective design of the Predi-COVID cohort study limits the risk of differential bias. The data collection process also ensures that the audio recordings were performed on the same day as the assessment of the symptoms, which limits the risk of reverse causation. This work relies on a large set of frequently reported symptoms in the literature.

   This work also has some limitations. First, our analysis only covers the discovery and internal validation phases of the vocal biomarker. Because the recordings were performed in real life, we have first cleaned and pre-processed the audio recordings and developed a pipeline to ensure that the vocal biomarkers training is as clean as possible, but we cannot completely rule out the possibility of having a few recordings of low quality. Potential sources of low quality of recordings include sub-optimal recording conditions in an uncontrolled environment, the use of lossy audio formats for data compression and the artifacts potentially introduced by noise reduction. There is currently no similar dataset existing on this topic with similar audio

recordings which prevents us from performing an external validation. Our vocal biomarker is mostly based on French and German voice recordings and as audio features may vary across languages or accents, our work will have to be replicated in other settings and populations [40]. The data used to train and test the models as well as the corresponding programs are open source and made available to the scientific community for replication or follow-up studies.

## Conclusions and perspectives

Using a simple, reproducible task of reading a standardized pre-specified text of 25 seconds, our work has demonstrated that it is possible to derive a vocal biomarker from a machine learning model to monitor the resolution of COVID-19 related symptoms with elevated accuracy and calibration. We have shown that voice is a non-invasive, quick, and cheap way to monitor COVID-19-related symptom resolution or aggravation. Such a vocal biomarker could be integrated into future telemonitoring solutions, digital devices, or in clinical practice for a rapid screening during a consultation to aid clinicians during anamnesis.

## Supporting information

**S1 File. Text to read for voice recording.**
(DOCX)

**S2 File. List of 66 acoustic features used based on OpenSmile_COMPARE 2016.**
(XLSX)

**S3 File. Density of LLD audio features for symptomatic vs asymptomatic cases—Android devices - 3gp audio format.**
(TIFF)

**S4 File. Density of LLD audio features for symptomatic vs asymptomatic cases—iOS devices—m4a audio format.**
(TIFF)

**S5 File. Selected features for the best model for each audio format.**
(XLSX)

## Acknowledgments

We thank the Predi-COVID participants for their involvement in the study, the members of the Predi-COVID external scientific committee for their expertise, as well as the project team, the IT team in charge of the app development, and the nurses in charge of recruitment, data and sample collection, and management on the field.

## Author Contributions

**Conceptualization:** Guy Fagherazzi, Abir Elbéji, Eduardo Higa, Markus Ollert, Petr V. Nazarov, Aurélie Fischer.

**Data curation:** Guy Fagherazzi, Lu Zhang, Eduardo Higa, Markus Ollert, Aurélie Fischer.

**Formal analysis:** Guy Fagherazzi, Aurélie Fischer.

**Funding acquisition:** Guy Fagherazzi, Markus Ollert, Aurélie Fischer.

**Investigation:** Guy Fagherazzi, Markus Ollert, Aurélie Fischer.

**Methodology:** Guy Fagherazzi, Lu Zhang, Abir Elbéji, Eduardo Higa, Vladimir Despotovic, Gloria A. Aguayo, Aurélie Fischer.

**Project administration:** Guy Fagherazzi, Markus Ollert, Aurélie Fischer.

**Resources:** Guy Fagherazzi, Markus Ollert, Gloria A. Aguayo.

**Supervision:** Guy Fagherazzi, Markus Ollert, Gloria A. Aguayo, Petr V. Nazarov, Aurélie Fischer.

**Validation:** Guy Fagherazzi, Abir Elbéji, Eduardo Higa, Gloria A. Aguayo, Aurélie Fischer.

**Visualization:** Guy Fagherazzi, Lu Zhang.

**Writing – original draft:** Guy Fagherazzi, Lu Zhang.

**Writing – review & editing:** Lu Zhang, Abir Elbéji, Eduardo Higa, Vladimir Despotovic, Markus Ollert, Gloria A. Aguayo, Petr V. Nazarov, Aurélie Fischer.

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
