## [Decision Letter · Decision Letter 0]

30 Mar 2022

PDIG-D-21-00115

A Voice-based Biomarker For Monitoring Symptom Resolution In Adults With COVID-19: Findings From The Prospective Predi-COVID Cohort Study

PLOS Digital Health

Dear Dr. Fagherazzi,

Thank you for submitting your manuscript to PLOS Digital Health. After careful consideration, we feel that it has merit but does not fully meet PLOS Digital Health's publication criteria as it currently stands. Therefore, we invite you to submit a revised version of the manuscript that addresses the points raised during the review process.

We look forward to receiving your revised manuscript.

Kind regards,

Ryan S McGinnis, Ph.D.

Academic Editor

PLOS Digital Health

Journal Requirements:

1. Please amend your detailed Financial Disclosure statement. This is published with the article, therefore should be completed in full sentences and contain the exact wording you wish to be published.

State the initials, alongside each funding source, of each author to receive each grant.

2. Please update your Competing Interests statement. If you have no competing interests to declare, please state: “The authors have declared that no competing interests exist.”

3. We ask that a manuscript source file is provided at Revision. Please upload your manuscript file as a .doc, .docx, .rtf or .tex. If you are providing a .tex file, please upload it under the item type ‘LaTeX Source File’ and leave your .pdf version as the item type ‘Manuscript’.

4. Please provide separate figure files in .tif or .eps format only and ensure that all files are under our size limit of 20MB.

Please also ensure that all files are under our size limit of 20MB.

For more information about how to convert your figure files please see our guidelines: https://journals.plos.org/digitalhealth/s/figures

Additional Editor Comments (if provided):

Reviewers' comments:

Reviewer's Responses to Questions

**Comments to the Author**

1. Does this manuscript meet PLOS Digital Health’s publication criteria? Is the manuscript technically sound, and do the data support the conclusions? The manuscript must describe methodologically and ethically rigorous research with conclusions that are appropriately drawn based on the data presented.

Reviewer #1: Yes

Reviewer #2: Yes

2. Has the statistical analysis been performed appropriately and rigorously?

Reviewer #1: Yes

Reviewer #2: Yes

Reviewer #3: Yes

3. Have the authors made all data underlying the findings in their manuscript fully available (please refer to the Data Availability Statement at the start of the manuscript PDF file)?

Reviewer #1: Yes

Reviewer #2: Yes

Reviewer #3: Yes

4. Is the manuscript presented in an intelligible fashion and written in standard English?

Reviewer #1: Yes

Reviewer #2: Yes

Reviewer #3: Yes

5. Review Comments to the Author

Reviewer #1: The paper presents a machine-learning based approach for the automatic identification of the presence of COVID-19 symptoms from the voice signal, proposing a “biomarker” which ultimately is the calibrated probability of being symptomatic. A pool of COVID-positive patients were recruited (Predi-COVID cohort), and vocal recordings were collected along with self-reported symptomatology questionaries, throughout a year-long period. There is heterogeneity of language and demographics within the subjects. A pipeline was built, starting from audio pre-processing followed by a wrapper-based feature selection and multiple classifiers, the better of which was chosen as the final model, whose calibrated outputs bring the “biomarker”. The test is independently divided between Android and iOS-based recording devices. 

The paper is well written and coherently organized, although there’s an abundance of sections. Moreover, the building and usage of a new dataset and the search for symptomatology are a strong contribution to the originality of the study. However, there is a number of weak points that affect the generality of the work. 

Especially when building custom datasets, which are often small, a peculiar attention to the cleanliness and a rigorous methodology should be employed. The recording conditions are not explained with sufficient detail. The main point that should be addressed with regards to the recordings delas with the source audio. In the “Data Analysis” section, the format of the files is stated to be 3GP for Android and M4A for iOS. Both of these extensions employ an AAC coding for audio signals, which is inherently lossy. This often leads to unsatisfactory and/or biased performances as well as loss of information for the audio analysis. An app is said to be used: since it manages the audio recording, we argue that it should record uncompressed/lossless audio. Additionally, no information is given on the compression parameters, sampling frequency and bit-depth of the collected audio. 

The automatic selection of the recordings is interesting, although very briefly explained in the “Pre-processing” section. Some points could be stated for the whole pre-processing and feature extraction part:

1) Some references or a concise discussion on the validity of the proposed selection methods could be beneficial, as it could be argued that “bad” recordings might have been left in if the selection methods weren’t sufficiently accurate. 

2) The pydub.effects.normalize function is stated to bring “a volume boost on the quiet parts”. Although the manual for the function states so, it is inherently wrong. This kind of normalization simply brings the peak of the signal to 0dbFS. The relative volume throughout different sections of the signal stays the same. Quiet parts are selectively raised in volume if compression is employed, or if normalization happens on smaller sections of the main signal, which appears not to be the case

3) Noise reduction is applied, using the logMMSE method. Using a noise reduction algorithm is a very delicate matter in audio analysis, as it inevitably changes and denatures the original data, possibly bringing artifacts not found in common speech signals, even when perceptual characteristics seem to improve. Therefore, a more thorough explanation and some proper referencing are needed in order to disclose the very algorithm used, the framework/library it has been employed with, and the reason why it should improve performance. 

4) It is not totally clear if there is a one-to-one correspondence between an audio recording of a patient and his questionaries. Does a patient record arbitrarily, whereas questionaries are (mostly) compulsory?

5) A better explanation behind the choice of adding the MFCC0 features, and on their nature as well, could be beneficial. 

The machine-learning part has some strong points, such as the use of many classifiers and steps, but needs some clarification and a more coherent explanation of some of the choices. 

In the Feature Selection section, the theory behind the RFE is not clearly explained, no references are present and the first sentence should be re-written. Additionally, it is not specified that it is a wrapper-based method and there is no clear explanation for the way the number of features is reached using Random Forest classifiers.

However, the most crucial point in the machine-learning section is the training-test split and the cross-validation throughout all the steps. What appears from the manuscript is that a 5-fold cross-validation was independently used within every different step, starting from the feature selection. However, since the 5 folds of in Classification step will be different, the features selected beforehand will be based on samples that may be the same that are now in the testing fold. This generates an inherent bias. Technically, the test data should never endure a feature selection, not even partly. 

Nevertheless, in section “Derivation of the digital biomarker” a “distribution in the test dataset” is mentioned. Which test dataset is it referring to? Perhaps the cross-validation happened only once, before the Feature Selection, as Figure 1 may suggest; this is not clear. 

Several other points should be clarified, justified or corrected: 

1) In the “Classification model selection and evaluation” it is said that “We evaluated the significance of the cross-validated scores of the final model with 1000 permutations.”. The sentence is unclear and the methodology should be explained. 

2) Since it is (understandably) stated as the most reliable indicator, a more thorough explanation would be needed for the MCC, as well as referencing. It would also be beneficial to present its formula, possibly in terms of false positives (FP), false negatives (FN), etc. Moreover, there is a “T” missing in Matthews’ name. 

3) The MCC is also the only indicator left out in the “Best predictive model” section.

4) At the end of the same section, the calibrations are deemed as “good” because of their Brier scores. However, such scores are not explained or referenced. A Brier score of 0.09 means a correct prediction with a 70% certainty, so what does “good” exactly mean in this scenario? 

The meta-analysis is also very hasty and lackluster. There is a brief digression on cough-based studies on COVID, and then it’s just stated that “no similar work to ours, i.e. focusing on individuals with COVID-19, has been previously reported. No comparison was possible with potential other models based on voice recordings rather than cough …”. However, this is definitely not true, as there is a handful of studies dealing with speech (sometimes associated with cough too) for COVID-19. The main ones that should be assessed are:

1) Robotti, C, Costantini, G et al. : Machine Learning-based Voice Assessment for the Detection of Positive and Recovered COVID-19 Patients (Journal of Voice, DOI:https://doi.org/10.1016/j.jvoice.2021.11.004) 

2) Pinkas, G, Karny, Y et al. : SARS-CoV-2 detection from voice. (IEEE Open J Eng Med. 2020)

3) Shimon, C, Shafat, G et al.: Artificial intelligence enabled preliminary diagnosis for COVID-19 from voice cues and questionnaires. (J Acoust Soc Am. 2021)

In the end, despite the novelty of the work and the undying necessity for quality research on COVID-19, we argue that this paper needs major revisions. A more thorough analysis and justification for the quality of the audio should be presented, as well as some clearance in the machine-learning section, and definitely a more extensive literature analysis.

Reviewer #2: Review of “a Voice-based Biomarker for Monitoring Symptom Resolution in Adults with Covid-19: Findings from the Prospective Predi-Covid Cohort Study” for PLOS

This paper sets out to derive a digital vocal biomarker to monitor COVID-19 symptom resolution. The authors used data from a prospective COVID study which included voice recordings from both symptomatic and asymptomatic participants who tested positive for COVID. Artificial intelligence models were trained to detect the resolution of symptoms.

The paper would be improved by including additional narrative clarifying the methodology. PLOS Digital Health includes studies from a variety of disciplines, and not everyone will have an extensive background in artificial intelligence modeling or handling digital voice recordings. Please include a list of the acronyms and their meanings.

Please include the total number of audio recordings and the number of symptomatic cases and asymptomatic cases in the text in the final paragraph under “methods.” 

In developing the list of COVID-19 related symptoms that were included in self-report questionnaires, did the authors consider “Flu-pro plus”? This is widely used in the United States.

On page 4 under the section “pre-processing, the sentence beginning “First, audio files…” appears to be missing something in parenthesis. The sentence on noise reduction appears to be incomplete.

On page 5, “Feature extraction,” please explain more about why you did this. This should be clear to those unfamiliar with sound processing and artificial intelligence modeling. The sentence starting “We compared the distribution…” talks about the arithmetic mean of symptomatic versus asymptomatic samples but also discusses separating android versus iOS audio in one confusing sentence. It would be clearer if these two concepts were described separately.

On page 5, “classification model selection and evaluation,” please explain the different models and why you would choose to test all four.

On page 7, in the last sentence of the section entitled “comparison with the literature,” do you mean “other” instead of the word “rather”?

On Table 1, does the table include only symptomatic participants? It’s confusing, because you mentioned you started out with 272 study participants, but I could not find the number of symptomatic versus asymptomatic participants.

On table 3, please include a key with a brief description of the machine learning algorithm, and the acronyms. On this table, it appears that “cases” means the number of voice recordings, not the number of actual participants. It would be good to clarify this.

In your discussion section, you might consider including possible future directions. You did not have a COVID free control group. I wondered whether the voices of people who had not tested positive for COVID would be different from people who tested positive but were asymptomatic. It appears that you compared voice recordings of participants indicating symptoms with voice recordings of participants who did not indicate any symptoms. So, over the course of time, would a person have been classified as “asymptomatic” following the resolution of symptoms that were reported earlier? If that is correct, it might be also interesting to look at the same participant’s voice while he/she indicates symptoms with that same participant’s voice after the symptoms have resolved.

Reviewer #3: This is a clearly written, timeliness paper. To my point of view, its major strengths rely on the good data availability and reproducibility, and the large specific covid-19 symptom database.

However, the paper have some major drawbacks, outlined next:

1. The database is large enough to perform state-of-the-art deep learning-based classification algorithms. However, only some traditional algorithms were tested. Although DL algorithms shouldn't necessary perform better, they should be included in the study.

2. The authors extract a huge number of features from OpenSmile (6472), and then they perform a feature selection using a Recursive Feature Elimination method. However, there is not detailed description of which voice features were extracted and selected, nor an analysis of the relevance of the extracted features for the classification. Are the best features more related to prosodic, only acoustic, dynamic features?

Such analysis would be crucial for a better understanding of a speech-based biomarker model.

3. Apart from the large database collection with its specificities, the current study does not represent a novelty with respect to the start of the art. The techniques used are not novel, and the discussion of what features are relevant as biomarker, which would be a good contribution to the paper, are not presented. The authors mention that "The most promising approach so far has been proposed by the MIT, as they achieved an elevated sensitivity (98.5% in symptomatic and 100% in asymptomatic individuals) and specificity (94.2% in symptomatic and 83.2% in asymptomatic individuals).". Which is the study they refer to? This approach is not properly referenced. Also, what are the main differences with respect to MIT's study that could make this contribution novel?

4. One of the limitations mentioned by the authors is that "Our vocal biomarker is mostly based on French and German voice recordings and as audio features may vary across languages or accents, our work will have to be replicated in other settings and populations." The overcome this, the experiments could have been performed separately by languages, so that it would be possible to analyse the language dependency to the biomarker.

Minor remark:

- "350ms"  "350 ms"

6. PLOS authors have the option to publish the peer review history of their article (what does this mean?). If published, this will include your full peer review and any attached files.

**Do you want your identity to be public for this peer review?** For information about this choice, including consent withdrawal, please see our Privacy Policy.

Reviewer #1: No

Reviewer #2: No

Reviewer #3: No

**Comments to the Author**

1. Does this manuscript meet PLOS Digital Health’s publication criteria? Is the manuscript technically sound, and do the data support the conclusions? The manuscript must describe methodologically and ethically rigorous research with conclusions that are appropriately drawn based on the data presented.

Reviewer #3: Yes

---

## [Decision Letter · Decision Letter 1]

13 Jul 2022

PDIG-D-21-00115R1

A Voice-based Biomarker For Monitoring Symptom Resolution In Adults With COVID-19: Findings From The Prospective Predi-COVID Cohort Study

PLOS Digital Health

Dear Dr. Fagherazzi,

Thank you for submitting your manuscript to PLOS Digital Health. After careful consideration, we feel that it has merit but does not fully meet PLOS Digital Health's publication criteria as it currently stands. Therefore, we invite you to submit a revised version of the manuscript that addresses the points raised during the review process.

Please submit your revised manuscript within 30 days Sep 11 2022 11:59PM. If you will need more time than this to complete your revisions, please reply to this message or contact the journal office at digitalhealth@plos.org. Please include the following items when submitting your revised manuscript:

We look forward to receiving your revised manuscript.

Kind regards,

Ryan S McGinnis, Ph.D.

Academic Editor

PLOS Digital Health

Journal Requirements:

Additional Editor Comments (if provided):

Reviewers' comments:

Reviewer's Responses to Questions

**Comments to the Author**

1. If the authors have adequately addressed your comments raised in a previous round of review and you feel that this manuscript is now acceptable for publication, you may indicate that here to bypass the “Comments to the Author” section, enter your conflict of interest statement in the “Confidential to Editor” section, and submit your "Accept" recommendation.

Reviewer #1: All comments have been addressed

Reviewer #2: All comments have been addressed

2. Does this manuscript meet PLOS Digital Health’s publication criteria? Is the manuscript technically sound, and do the data support the conclusions? The manuscript must describe methodologically and ethically rigorous research with conclusions that are appropriately drawn based on the data presented.

Reviewer #1: Yes

Reviewer #2: Yes

3. Has the statistical analysis been performed appropriately and rigorously?

Reviewer #1: Yes

Reviewer #2: Yes

4. Have the authors made all data underlying the findings in their manuscript fully available (please refer to the Data Availability Statement at the start of the manuscript PDF file)?

Reviewer #1: Yes

Reviewer #2: Yes

5. Is the manuscript presented in an intelligible fashion and written in standard English?

Reviewer #1: Yes

Reviewer #2: Yes

6. Review Comments to the Author

Reviewer #1: MINOR REVISION

The authors addressed most of the concerns with reasonable answers. 

Several limitations on the scientific validity of the proposed study still emerge, especially on the quality of the proposed dataset and on the reproducibility of the methodology. However, the proposal is coherent and interesting enough for publication, with a suggested minor revision addressing the following points:

- In the response, the authors state that lossy audio is “a necessary trade-off in anticipation of the future implementation in real life of such a digital health solution in practice where multiple devices are used”. We politely disagree, as losing on audio quality on sources as diverse and complex as self-recorded voice could really hinder the quality of the measurements; on the other hand, it is feasible to provide an App or API that allows different devices to record lossless audio.

- For the abovementioned reasons, it would be beneficial to clearly state the main limitations of this study for the readers to realize: lossy audio, sub-optimal recording conditions and lack of control (already mentioned in the Limitations section), non-homogeneous processing due to the .m4a compression and possible artifacts induced with the usage of logMMSE noise reduction. 

- The explanation of the 1000 permutations for training models is still unclear: which “targets” are permuted? Please re-write it

- It is unclear to me why the results of the proposed references detecting COVID-19 from the voice are not comparable to those in the present study. “Symptomatic” COVID is of course the main aim when trying to pre-diagnose it from the voice. However, this is just a minor consideration.

We thank the authors for following our suggestions.

Reviewer #2: Excellent contribution. Thank you for your response to reviewer comments.

7. PLOS authors have the option to publish the peer review history of their article (what does this mean?). If published, this will include your full peer review and any attached files.

**Do you want your identity to be public for this peer review?** For information about this choice, including consent withdrawal, please see our Privacy Policy.

Reviewer #1: None

Reviewer #2: No

---

## [Editor Report · Decision Letter 2]

26 Aug 2022

A Voice-based Biomarker For Monitoring Symptom Resolution In Adults With COVID-19: Findings From The Prospective Predi-COVID Cohort Study

PDIG-D-21-00115R2

Dear Dr Fagherazzi,

We are pleased to inform you that your manuscript 'A Voice-based Biomarker For Monitoring Symptom Resolution In Adults With COVID-19: Findings From The Prospective Predi-COVID Cohort Study' has been provisionally accepted for publication in PLOS Digital Health.

Best regards,

Ryan S McGinnis, Ph.D.

Academic Editor

PLOS Digital Health